# A Two-stage Segmentation Neural Network for MICCAI FLARE22 Challenge

Jianping Zhang[1] and Shuhui Jia[2]

Xiangtan University, Xiangtan City, China

**Abstract.** Abdominal organ segmentation plays an important role in medical image processing. In this work, our goal is to segment thirteen organs of the abdomen in a semi-supervised way. We apply the attention block to the DMFNet, and propose a new Attention DMFNet for medical imaging, which can automatically learn and focus target structures of different shapes and sizes. The DMFNet is a highly efficient 3D CNN, which can realize real-time dense volume segmentation. It uses 3D multi fiber units composed of lightweight 3D convolution network to significantly reduce the computational cost. Models trained with attention blocks implicitly learn to suppress irrelevant regions in an input image while highlighting salient features useful for a specific task. Integrating attention blocks into the DMFNet can improve model sensitivity and prediction accuracy with minimal computational overhead. And we adopt a two-stage approach. In the first stage, the foreground containing all organs is segmented. In the second stage, thirteen organs are segmented on the basis of the first stage. We use labeled data to train the teacher model, use the teacher model to predict the unlabeled data, and take the segmentation result as pseudo labels for the following training. And then the data with true labels and the data with pseudo labels are used to train student models with the help of robust loss functions, namely beta cross-entropy, symmetric cross-entropy, and generalized cross-entropy. Finally, the trained student model is used to predict the data.

**Keywords:** Segmentation · Pseudo Supervision · Attention Block.

## 1 Introduction

There are three major challenges in this task. The first challenge comes from the diversity of the dataset, which including multi-center, multi-phase, multi-vendor, and multi-disease cases. Medical image segmentation has not been solved today. An important reason is the complexity and diversity of medical images. Due to the differences in the imaging principle of medical image and the characteristics of tissue itself, the formation of image is affected by noise, field offset effect, local volume effect and tissue motion. Compared with ordinary image, medical image inevitably has the characteristics of fuzziness and non-uniformity. In addition, the anatomical structure and shape of human body are complex, and there are considerable differences between people. All these bring difficulties to medical

image segmentation. Traditional segmentation techniques either fail completely or require some special processing techniques. Therefore, it is necessary to study image segmentation methods in the field of medical application.

The second challenge comes from the efficiency requirement for the proposed solutions. Medical image data are mostly three-dimensional data with large size, especially abdominal medical CT images. Moreover, the existing three-dimensional convolutional neural networks generally have the disadvantages of large parameters and high requirements for GPU. This is very disadvantageous to the clinical application of the convolutional neural network.

The third problem is that there are few labeled data and many unlabeled data in the data used for training, and there are great differences between them, which has more stringent requirements on the segmentation ability of the model.

As for the first challenge, we use a robust loss function to solve the problem of large data differences. In order to solve the second difficulty, we use the DMFNet as the solution, and add attention blocks to improve the segmentation accuracy. The reasons are as follows. First, the DMFNet has less parameters and it is fast. Second, attention blocks can identify salient image regions and prune feature responses to preserve only the activations relevant to the specific task. Third, it shows powerful performance on several segmentation tasks. On the third question, we use semi-supervised method to train the model to make better use of unlabeled data. Specifically, we first use labeled data to train the teacher model, and then predict the unlabeled data, and take its prediction results as pseudo labels. Finally, the data, the true labels and the pseudo labels are sent into the student model for training, so as to improve the segmentation ability of the model for all data.

## 2   Method

A detail description of the method used, a schematic representation of the method is recommended.

### 2.1   Preprocessing

Full description of any pre-processing strategy, how the data is normalized. Please details the following aspects

- Cropping strategy
- Resampling method for anisotropic data
- Intensity normalization method
- Others

### 2.2   Proposed Method

Motivation and description of the method details. **Pre-trained models are not allowed to use in this challenge.**

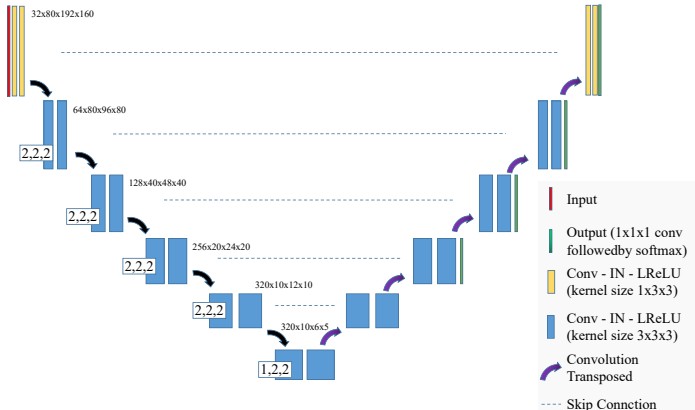

**Fig. 1.** Network architecture

Please provide a figure to show your network architecture. Figure 1 illustrates the applied 3D nnU-Net [4], where a U-Net architecture is adopted.

Strategies to use the unlabelled cases

Network architecture details (Based on the winning solutions in FLARE 2021, we recommend using two-stage framework)

Loss function: we use the summation between Dice loss and cross entropy loss because compound loss functions have been proved to be robust in various medical image segmentation tasks [5].

Strategies to improve inference speed and reduce resource consumption (Based on the winning solutions in FLARE 2021, we recommend using ONNX or TensorRT to speed up inference process)

### 2.3   Post-processing

*Description of post-processing of the model outputs to get the final output in training stage.*

## 3   Experiments

### 3.1   Dataset and evaluation measures

The FLARE2022 dataset is curated from more than 20 medical groups under the license permission, including MSD [7], KiTS [2,3], AbdomenCT-1K [6], and TCIA [1]. The training set includes 50 labelled CT scans with pancreas disease and 2000 unlabelled CT scans with liver, kidney, spleen, or pancreas diseases. The validation set includes 50 CT scans with liver, kidney, spleen, or pancreas diseases. The testing set includes 200 CT scans where 100 cases has liver, kidney,

spleen, or pancreas diseases and the other 100 cases has uterine corpus endometrial, urothelial bladder, stomach, sarcomas, or ovarian diseases. All the CT scans only have image information and the center information is not available.

The evaluation measures consist of two accuracy measures: Dice Similarity Coefficient (DSC) and Normalized Surface Dice (NSD), and three running efficiency measures: running time, area under GPU memory-time curve, and area under CPU utilization-time curve. All measures will be used to compute the ranking. Moreover, the GPU memory consumption has a 2 GB tolerance.

### 3.2   Implementation details

**Environment settings**  The development environments and requirements are presented in Table 1.

**Table 1.** Development environments and requirements.

| Windows/Ubuntu version | Ubuntu 18.04.5 LTS |
|---|---|
| CPU | Intel(R) Core(TM) i9-7900X CPU@3.30GHz |
| RAM | 16×4GB; 2.67MT/s |
| GPU (number and type) | Four NVIDIA V100 16G |
| CUDA version | 11.0 |
| Programming language | Python 3.9 |
| Deep learning framework | Pytorch (Torch 1.10, torchvision 0.2.2) |
| Specific dependencies | |
| (Optional) Link to code | |

**Training protocols**  Please describe at least the following aspects:

Data augmentation (Based on the winning solutions in FLARE 2021, we recommend using extensive data augmentation) patch sampling strategy, optimal model selection criteria

## 4   Results and discussion

Note: Please describe at least the following aspects:
The effect of using unlabelled cases;
What kind of cases the proposed method works well?
What are the possible reasons for the failed cases or organs?
Segmentation efficiency analysis

### 4.1   Quantitative results on validation set

Currently, you can report the Dice score on validation set
Please do ablation study to analysis the effect of unlabelled data.

**Table 2.** Training protocols.

| | |
|---|---|
| Network initialization | "he" normal initialization |
| Batch size | 2 |
| Patch size | 80×192×160 |
| Total epochs | 1000 |
| Optimizer | SGD with nesterov momentum ($\mu = 0.99$) |
| Initial learning rate (lr) | 0.01 |
| Lr decay schedule | halved by 200 epochs |
| Training time | 72.5 hours |
| Number of model parameters | 41.22M[1] |
| Number of flops | 59.32G[2] |
| $CO_2$eq | 1 Kg[3] |

**Table 3.** Training protocols for the refine model (if using two-stage framework).

| | |
|---|---|
| Network initialization | "he" normal initialization |
| Batch size | 2 |
| Patch size | 80×192×160 |
| Total epochs | 1000 |
| Optimizer | SGD with nesterov momentum ($\mu = 0.99$) |
| Initial learning rate (lr) | 0.01 |
| Lr decay schedule | halved by 200 epochs |
| Training time | 72.5 hours |
| Number of model parameters | 41.22M[4] |
| Number of flops | 59.32G[5] |
| $CO_2$eq | 1 Kg[6] |

## 4.2   Qualitative results on validation set

This part is optional during validation phase since you do not have validation ground truth.

## 4.3   Segmentation efficiency results

## 4.4   Limitation and future work

## 5   Conclusion

The main finding and results

**Acknowledgements** The authors of this paper declare that the segmentation method they implemented for participation in the FLARE 2022 challenge has not used any pre-trained models nor additional datasets other than those provided by the organizers. The proposed solution is fully automatic without any manual intervention.

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
