# OpenReview forum: "A Two-stage Segmentation Neural Network for MICCAI FLARE22 Challenge"
_MICCAI.org/2022/Challenge/FLARE_

### Official Review · Reviewer_WgfD · 2022-09-16
**It is bad and miss a lot of information.**

**Rating:** 2
**Confidence:** 5

**Review:**

This paper missing the details of method,result and the conclusion.
It seemed that this paper is just the official provided template.

---

### Official Review · Reviewer_sEGU · 2022-09-16
**The paper is not finished**

**Rating:** 2
**Confidence:** 1

**Review:**

Strength: The authors clear figured out the core difficulty of this segmentation dataset.
Weakness: This paper is totally unfinished, many core parts are not written.

---

### Official Review · Reviewer_zUpm · 2022-09-17
**The structure of the paper is incomplete.**

**Rating:** 1
**Confidence:** 1

**Review:**

This paper does not describe the methods and results in detail. The paper is incomplete, missing the content of methods and results.

---

### Official Review · Reviewer_ndSy · 2022-09-19
**A two-stage neural network for medical image segmentation**

**Rating:** 2
**Confidence:** 5

**Review:**

The authors apply the attention block to the DMFNet and propose a new Attention DMFNet for medical imaging, which can automatically learn and focus target structures of different shapes and size.

But this article lacks all necessary parts.This article is a official template except for the summary and introduction.
Please confirm whether the wrong file has been uploaded?

---

### Official Review · Reviewer_59Gh · 2022-09-19
**The paper is incomplete and needs further explanation.**

**Rating:** 2
**Confidence:** 4

**Review:**

Pros:
1.The analysis of the challenges were given in introduction.

Cons:
1.The paper is incomlete, and the details of the pipeline remain unclear.
2.The concept of 'DMFNet' was only introduced in Abstract and Introduction section, but specific framework and principle was omit.

---

### Meta-Review · Program_Chairs · 2022-09-28

**Recommendation:** Major Revision
**Confidence:** 5

**Metareview:**

Reviewers raise many concerns and suggestions. Please address all comments in the revised manuscript.